# Influence of Showerhead–Sample Distance (GAP) in MOVPE Close Coupled Showerhead Reactor on GaN Growth

**DOI:** 10.3390/ma12203375

**Published:** 2019-10-16

**Authors:** Robert Czernecki, Karolina Moszak, Wojciech Olszewski, Ewa Grzanka, Mike Leszczynski

**Affiliations:** 1Institute of High Pressure Physics, Polish Academy of Sciences, Sokolowska 29/37, 01-142 Warsaw, Poland; elesk@unipress.waw.pl (E.G.); mike@unipress.waw.pl (M.L.); 2Lukasiewicz Research Network – PORT Polish Center for Technology Development, Stablowicka 147, 54-066 Wroclaw, Poland; Karolina.Moszak@port.org.pl (K.M.); Wojciech.Olszewski@port.org.pl (W.O.)

**Keywords:** nitrides, epitaxy, metalorganic vapour phase epitaxy

## Abstract

The distance between the showerhead and the sample surface (GAP) is one of the main growth parameters of the commonly used research reactor, Close Coupled Showerhead. We examine its influence on the growth rate of GaN layers deposited under various conditions (growth temperature, carrier gas, V/III ratio and growth pressure). Regardless of other growth parameters, increasing the GAP value leads to a reduction in the growth rate.

## 1. Introduction

Metalorganic Vapour Phase Epitaxy (MOVPE) is the most common technique applied for growing epitaxial structures used in electronics and optoelectronics. However, MOVPE growth models are still far from satisfactory, because in this technique we deal with a large number of growth parameters (such as: temperature, pressure, carrier gas) which are not independent to each other. Additionally, growth is strongly influenced by the construction of the reactor.The most popular research MOVPE reactor, Close Coupled Showerhead (CCS) is manufactured by AIXTRON [1]. In this reactor, the reactant gases are introduced through a large number of small holes. The showerhead is watercooled, and its temperature is around 50 ${}^{\circ }$C. The resistance-heated susceptor, with the sample on it, is installed very close to the showerhead; typical distance is 11 mm. At such short distance (GAP), there is a huge temperature gradient (around 1000 deg). Moreover, small GAP creates a problem of deposits on the showerhead and the “memory” of the reactor. This issue was discussed in [2,3,4,5]. The easiest way to overcome these disadvantages is to increase GAP, however, as mentioned, this new growth parameter is not independent to the others.

In this paper, we report on the experimental results of the GaN growth rate as a function of GAP for different temperature, pressure and flows of N${}_{2}$, H${}_{2}$, NH${}_{3}$. Such results should be helpful in optimizing the growth conditions in the CCS reactors.

## 2. Experimental

The GaN epi layers were deposited in Aixtron CCS MOVPE reactor with a 3 × 2 inch wafer configuration. This reactor is additionally equipped with a distance regulation system between the sample surface and the showerhead (GAP adjustment) from 5 to 25 mm (default GAP value is 11 mm). As substrates, we used 3 μm thick GaN templates grown on sapphire with standard off-cut angle beetween 0.3–0.35 deg. along m-plane. Trimethylgallium (TMGa) and ammonia (NH${}_{3}$) were used as gallium and nitrogen precursors, respectively. Hydrogen (H${}_{2}$) or nitrogen (N${}_{2}$) were used as carrier gas during the layer growth. We used total flow = 8 standard liter per minute (slm). Substrate temperature was controlled by a thermocouple placed below the graphite susceptor. The growth rate of the GaN layer was determined using a laser reflectometer with a laser wavelength of 635 nm.

## 3. Results

One of the main effects of changing the GAP distance is the modification of the real surface temperature of the substrate. Figure 1 shows how this value changes for different carrier gases (H${}_{2}$ and N${}_{2}$) for a fixed thermocouple value (1140 ${}^{\circ }$C).

Real surface temperature was determined using the Laytec EPI TT system [6]. As can be seen, the real surface temperature of the substrate in the H${}_{2}$ atmosphere is always lower than for the case than for N${}_{2}$ used as carrier gas and changes slightly throughout the whole GAP range regardless of the pressure inside the reactor. This is due to the difference in thermal conductivity between H${}_{2}$ and N${}_{2}$.

Figure 2 shows how the growth rate of undoped GaN layers (uGaN) grown at 150 mbar changes for three different growth temperatures.

As can be seen, for high growth temperature (1030 ${}^{\circ }$C) the growth rate does not depend on the carrier gas used. For a typical growth temperature (950 ${}^{\circ }$C), it can be seen that for small GAP values (below 18 mm) the growth rate in H${}_{2}$ atmosphere is about 20% higher, what in the literature [7] is explained by the more effective decomposition of TMGa. For a low growth temperature (770 ${}^{\circ }$C) and the highest GAP value (24 mm), the growth rate in the N${}_{2}$ atmosphere is higher than in H${}_{2}$, what is very unusual.

Figure 3 shows how the growth rate of undoped GaN layers (uGaN) grown at 950 ${}^{\circ }$C changes for three different growth pressures.

As you can see, regardless of the carrier gas used, the growth rate decreases significantly as the pressure in the reactor increases. The largest decrease is observed for the maximum GAP value. It should be assumed that increasing the pressure increases the thickness of the “boundary layer” between the gas phase and the solid state through which the reagents diffuse into the surface of the substrate [8].

Another possible explanation of decreasing growth rate for high GAP value may be gas-phase parasitic reactions. Figure 4 shows how the growth rate changes in the GAP function for different V/III ratios changed by NH${}_{3}$ flow (at constant total flow).

As can be seen, when N${}_{2}$ is used as the carrier gas, the influence of NH${}_{3}$ flow is practically negligible. In case when H${}_{2}$ is used as a carrier gas, decreasing NH${}_{3}$ flow (lower V/III ratio) leads to an increase in growth rate, which can prove a smaller amount of parasitic reactions in the gas phase between NH${}_{3}$ and TMGa. Greater profit from reducing the amount of NH${}_{3}$ in the gas phase can be seen for higher GAP values.

## 4. Conclusions

We presented experimental results regarding GaN growth rate as a function of GAP for various temperatures, pressure, type of carrier gas and V/III ratio. In all cases, the increase of GAP resulted in a decrease of growth rate. The strongest effect of GAP value was at high pressure, what indicates gas phase parasitic reactions and, probably, because of the thicker/denser “boundary layer” above the substrate.

## Figures and Tables

**Figure 1 materials-12-03375-f001:**
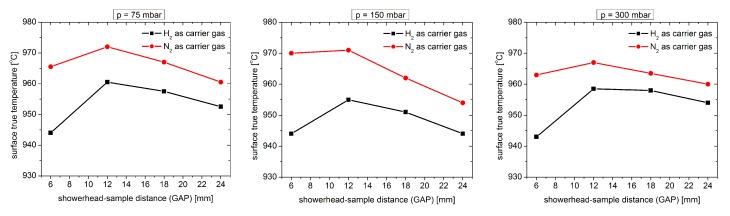
Surface temperature as a showerhead-sample distance (GAP) function for a fixed thermocouple value (1140 ${}^{\circ }$C) for different pressures inside the reactor.

**Figure 2 materials-12-03375-f002:**
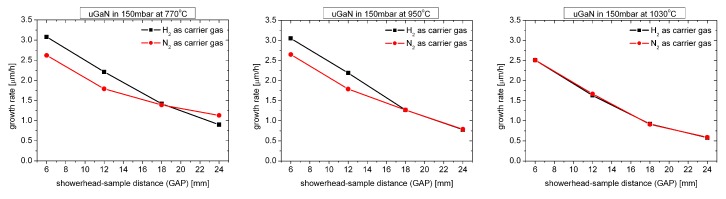
Growth rate of uGaN layers as a function of GAP distance for three different growth temperatures.

**Figure 3 materials-12-03375-f003:**
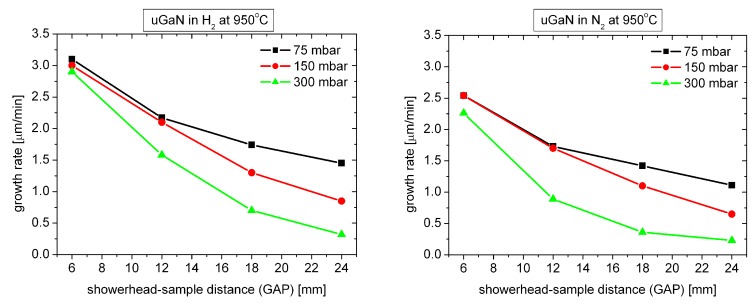
Growth rate of uGaN layers as a function of GAP distance for three different growth pressures.

**Figure 4 materials-12-03375-f004:**
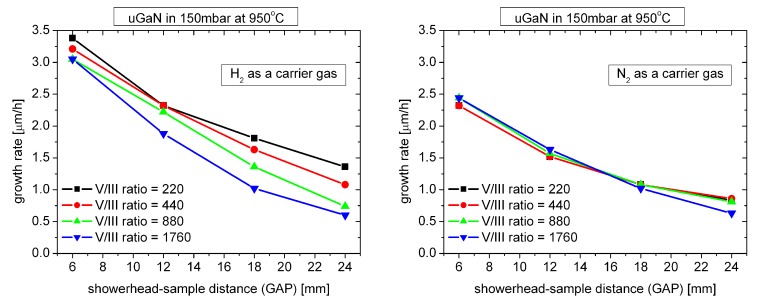
Growth rate of uGaN layers as a function of GAP distance for different V/III ratios (obtained for different NH${}_{3}$ flows).

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
