# Peer review of "Influence of Showerhead–Sample Distance (GAP) in MOVPE Close Coupled Showerhead Reactor on GaN Growth"

_materials, 2019, doi:10.3390/ma12203375_

Round 1
Reviewer 1 Report
GaN films are a promising material for optoelectronics. Improving of the epitaxial layers quality is crucial for practical application. The physicochemical processes occurring during the growth of these GaN films are poorly studied.
This work has a technological focus, and is associated with the study of the influence of GAP, temperature, carrier gas, composition and others on the speed of layer growth.
I note that the manuscript does not contain a list of references. This does not allow us to fully appreciate the completeness of the information in the introduction and in the discussion section.
What substrate orientation was used? The issues of structure perfection of layers and their electrophysical parameters depending on the parameters studied are not considered in the work.
The manuscript needs substantial revision.
Author Response
Dear Sir / Madam,
Thank you very much for valuable comments on my article. I would like to apologize for mistakenly omitting references in the first version of the work. I have added information about off-cut angle of sapphire substrates.
Best regards,
Robert Czernecki
Reviewer 2 Report
Please add a list of references and complete the main text of the manuscript: the references are given as [? ] and the figures referred to in the text as "figure ??".
Author Response
Dear Sir / Madam,
Thank you very much for valuable comments on my article.I would like to apologize for omitting the references in the first version by mistake and complete in the next version.
Best regards,
Robert Czernecki
Reviewer 3 Report
Affiliations is missing/wrong References are missing some minor misspelling error (intention instead on intension)
Author Response
Dear Sir/Madam,
Thank you very much for valuable comments on my article. I would like to apologize for omitting the references in the first version by mistake and complete in the next version. In the revised version, I improved the introduction and corrected the affiliations.
Best regards,
Robert Czernecki
Reviewer 4 Report
The current work by Czernecki and coworkers essentially represents an incomplete technical report of a negative result. The authors have hypothesized that by increasing the distance between the showerhead and the sample surface in CCS-MOVPE reactors they will avoid deposition on the showerhead, however they seemingly do not achieve this but observed slower deposition rate. The authors speculate the cause why the experiment is not successful; however, it is also questionable to what degree the outcome could have been overseen. The authors do not display any reference which significantly hurts the rhetoric and the scholarship of the current work. Inconsistencies in the text markings, incorrectly displayed labeling for the images render the reading of the text difficult.
The overall text is also extremely short, probably aiming the format of a communication, however it does not demonstrate urgency nor self-sufficient comprehensive scholarship to justify this. I am also not against presentation of negative results, but they should be placed in a context of the current understanding of the field and how the new work resolves existing misconceptions. The current manuscript does not provide this and in such a form I cannot recommend it for publication.
Author Response
Dear Sir/Madam,
Thank you very much for valuable comments on my article. I would like to apologize for omitting the references in the first version by mistake and complete in the next version. In the revised version, I improved the introduction. Our research presents only experimental data, but we are working on a theoretical model explaining these data. In the literature I have not found any work investigating the impact of GAP value on GaN growth rate.
Best regards,
Robert Czernecki
Round 2
Reviewer 1 Report
Unfortunately, the authors did not take into account
all the comments made earlier. The influence of technological factors on the crystalline perfection
of the grown GaN layers would significantly improve the quality of this
investigation, because improving the layers quality for practical
application is the main task. I hope the authors will be able to carry out this study in the future.
Reviewer 4 Report
The technical presenting style of the work by Dr. Czernecki and coworkers has been improved since the previous version. The references are in place as well as the figure indicators in the main text.
In my view, the content, the novelty and the overall scientific rigor does not meet the standards and expectations for comprehensive manuscript published in Materials or other journals. However, I still think that this is a solid technical report of a negative result. It deserves to be published, but other MDPI platforms such as Reports may be more suitable for that.